# The Discrepancy between Subjective and Objective Evaluations of Cognitive and Functional Ability among People with Schizophrenia: A Systematic Review

**DOI:** 10.3390/bs14010030

**Published:** 2023-12-30

**Authors:** Molly Harris, Emily Blanco, Hunter Howie, Melisa Rempfer

**Affiliations:** 1Department of Psychology, University of Missouri-Kansas City, Kansas City, MO 64110-2499, USA; 2Department of Psychology, University of Kansas, Lawrence, KS 66045, USAmrempfer@ku.edu (M.R.)

**Keywords:** self-report, accuracy, neuropsychological assessment, functional capacity, depression, cognition, recovery

## Abstract

Background: An important aspect of recovery in schizophrenia relates to one’s subjective, lived experience. Self-report is a subjective measurement method with yet-uncertain utility in the assessment of functioning among individuals diagnosed with schizophrenia-spectrum disorder. No review to date has comprehensively synthesized existing research to evaluate the degree of correspondence, or lack thereof, between subjective and objective assessments of cognitive and everyday functioning, nor how extant data can inform the use of self-reported information in treatment and research. Methods: A systematic review was completed to provide a broad perspective of the literature on this topic. Relevant manuscripts were identified via a search strategy using key terms in PubMed and PsycINFO and a review of manuscript bibliographies. Twenty-six studies met the inclusion criteria. Results: These studies show minimal to modest associations between subjective assessments of cognition and everyday functioning and objective assessments of these domains, including informant reports and neuropsychological and behavioral measures. Individuals with schizophrenia appear to overestimate their functioning when compared to objective measures. Depression and greater cognitive ability tend to predict greater correspondence between subjective and objective assessments of cognition and everyday functioning. Discussion: This review discusses how we might understand the low correspondence between subjective and objective measures of functioning and provides recommendations for using and eliciting self-reported information in the pursuit of recovery-centered practices.

## 1. Introduction

There are various definitions of recovery in schizophrenia, reflecting not only clinical and functional outcomes but also personal and subjective perspectives that capture individuals’ lived experiences. Self-report measures of functioning are used in research and clinical settings because they capture lived experience, are cost-effective, and are easy to administer. However, subjective reporting among people with schizophrenia is also controversial because self-report measures frequently do not correspond with “objective” measures [1,2,3]. This phenomenon is typically attributed to a lack of self-awareness or low insight into symptoms, cognitive difficulties, and functional deficits [4,5]. Many studies have explored discrepancies between self-report measures and measures derived from clinician ratings, informant reports, or behavioral and neuropsychological assessments. A concern raised by this research is that self-report measures may not be valid metrics of true cognition and functioning and should be used with caution [6,7].

Patient-reported outcomes (PROs) are becoming more central to the evaluation of services in mental health and other healthcare settings, as they indicate quality of care and consumer satisfaction [8,9]. Beyond assessing treatment programs, PROs have been shown to be predictive of important clinical outcomes, including long-term functioning [10] and rehospitalization [11]. These studies suggest that subjective assessments can have clinical significance. A deeper understanding of subjective reporting among individuals with schizophrenia is necessary to facilitate both clinical assessment and person-centered, recovery-oriented care. Though the reliability of self-reported functioning and PROs among people with schizophrenia has been called into question, there is no consensus on how widespread inaccurate self-report is, whether inaccuracy can be reliably predicted, or how to approach the integration of discordant reporting into clinical care and assessment. This review aims to provide a systematic account of the validity of self-reported information in both cognitive and functional domains among people with schizophrenia. We further review the characteristics of individuals that may predict higher convergence of subjective reporting with objective measures and discuss future directions for this body of research.

## 2. Method

### 2.1. Search Strategy

The search strategy was based on the Preferred Reporting Items for Systematic Reviews and Meta-Analyses (PRISMA) guidelines [12]. This review was not pre-registered prior to data collection. A search was performed using the PubMed and PsycINFO databases for peer-reviewed journal articles published before April 2023. We additionally searched the reference lists of all included articles to further identify eligible studies. The following search string was applied for electronic searches of databases: (“self-report” OR “self-assessment” OR “self-rating” OR “self-monitoring” OR “self-evaluation” OR “self-appraisal” OR “self-awareness” OR “self-perceived” OR “consumer reported outcomes” OR subjective OR metacognitive) AND (accuracy OR discrepancy OR convergence OR validity) AND (“schizophrenia”). These keywords were determined via collaboration among all authors. After removing duplicate records, the electronic search yielded 2372 records whose abstracts were reviewed, and 96 articles were selected for further review based on information found in the abstracts. Twenty-two articles were identified for inclusion based on the criteria below. The bibliographies of all included articles were manually searched for additional material, and four articles were identified for inclusion. In total, 26 articles were included in this review. A flowchart of the included articles can be found in the Appendix A.

Two researchers (M.H. and E.B.) independently reviewed the titles and abstracts of records identified through the electronic database search and selected studies for full-text screening based on our eligibility criteria. Then, three researchers (M.H., E.B., and H.H.) independently screened full-text studies to determine eligibility for inclusion. Disagreements or uncertainty about eligibility were resolved through discussion until a consensus was reached.

### 2.2. Inclusion Criteria

Studies must include samples of only individuals diagnosed with schizophrenia spectrum disorders (including schizophrenia, schizoaffective disorder, delusional disorder, and all other forms of psychotic disorders); studies that included groups of individuals with other conditions (e.g., healthy control groups or groups of individuals with other diagnoses) where the group data were analyzed and reported separately were eligible for inclusion;Studies must use a self-report measure of cognitive or functional ability or achievement (i.e., outcomes related to everyday functioning and community involvement);Studies must use an informant or performance-based measure of the same cognitive or functional ability that was measured by self-report;Studies must statistically compare the self-report measure to its complementary informant or performance measure;Participants must be over the age of 18.

### 2.3. Exclusion Criteria

Systematic reviews, meta-analyses, and book chapters were not eligible for inclusion;Articles written in languages other than English where no published English translation is available;Studies utilizing mixed samples that include diagnoses other than schizophrenia spectrum disorders where data for participants with schizophrenia spectrum disorders were not analyzed and reported separately.
We did not restrict studies based on study location, sample size, or study design. We considered studies that utilized a range of measures for cognitive and functional ability, including surveys or questionnaires completed by participants, informants, or clinicians; performance-based measures of any neurocognitive ability, such as memory, executive function, or processing speed; and performance-based measures of functional ability or everyday functional achievement, such as the ability to perform activities of daily living.

Studies that did not directly compare subjective and objective measures of cognitive or functional ability were not included in this review. Further, we did not include analyses of social cognition as this is considered a separate construct from neurocognitive ability [13]. However, studies that analyzed social cognition as well as neurocognitive or functional ability were eligible for inclusion, though we report only the results that pertain to neurocognitive or functional ability.

### 2.4. Data Extraction and Synthesis

The full articles were read by three independent reviewers (M.H., E.B., and H.H.). Reviewers summarized the sample size, measures, reported outcomes, and findings of each article using a standardized data extraction form. All reviewers compared extracted data as a group, and any discrepancies were resolved through discussion. Due to high heterogeneity in study design and instrumentation across studies, a meta-analysis could not be performed. Instead, the results of the included studies are summarized in a narrative form. Extracted data, including estimates of effect sizes when available, key findings, and study characteristics, are reported in Table 1, Table 2 and Table 3.

For each study, we sought to include any estimates of the size of the discrepancy between subjective and objective data, such as mean differences, correlations, or discrepancy scores. Information about the percentage of a study sample that was classed as “inaccurate” estimators was also included. When available, variables used to predict the discrepancy between subjective and objective sources were extracted, and we sought to report their effect. Additional data, such as participant demographics, are included in Table 1, Table 2 and Table 3.

A number of papers included in this review were reports from larger studies and, therefore, used similar samples of participants. Given that additional reports provided unique and useful information, their findings are discussed separately. However, papers that disclosed or appeared to use the same dataset are marked in Table 1, Table 2 and Table 3.

### 2.5. Risk of Bias Assessment

To evaluate the risk of bias for each included study, we utilized the Risk of Bias Utilized for Surveys Tool (ROBUST) [39], a generic tool developed to assess the quality of studies without interventions. This tool includes eight binary items (“yes” or “no”) that pertain to criteria such as sampling procedures (i.e., whether researchers used random sampling or not), reliability of measures (i.e., whether researchers reported adequate reliability statistics), and data management procedures (i.e., whether data were cleaned prior to analysis). The specific items can be found in the Appendix A. All authors (MH, EB, JHH, and MR) independently assessed a random subset of five articles and compared findings to establish appropriate criteria for this review. The remaining articles were independently assessed by three authors (M.H., E.B., and H.H.), and any disagreements were resolved via discussion. Total scores ranged from 0–8, with lower scores indicating a greater risk of bias. We classified studies that scored 0–3 as “high risk” and studies that scored 7–8 as “low risk.” Overall, risk of bias scores ranged from 2–6 for the studies included in this review. Six reports were classified as high risk, while 20 fell in the medium risk range.

## 3. Results

### 3.1. Self-Reported Cognition

Four studies used the Subjective Scale to Investigate Cognition in Schizophrenia (SSTICS) to explore the accuracy of self-reported cognitive functioning. Lecardeur et al. [17] compared the SSTICS and the Frankfurt-Pamplona Subjective Experiences Scale (FPSES) to the cognition factor of the Positive and Negative Symptoms Scale (PANSS). They found a moderate correlation between the SSTICS and PANSS cognition, but the FPSES failed to correlate with the PANSS. Using a translated version of the SSTICS, Johnson and colleagues [18] compared self-reported cognition to a neuropsychological battery and found no significant correlations between the SSTICS total score and any of the neuropsychological tests in their study. Similarly, Sellwood et al. [20] examined correlations between the SSTICS and the Brief Assessment of Cognition in Schizophrenia (BACS), but no significant correlations among the SSTICS subscales and the BACS domain scores were found.

Prouteau et al. [22] split participants with schizophrenia into two subgroups based on their performance on the Trail Making Test (TMT): a subgroup identified as “executively normal” and a “dysexecutive” subgroup. Researchers then calculated discrepancy scores using the SSTICS and performance on the Modified Card Sorting Test (MCST) to represent neurocognitive insight and compared the two subgroups to each other and a control group. The schizophrenia subgroups (dysexecutive and executively normal) did not differ from each other in neurocognitive insight scores, nor did the executively normal group differ from the control group. However, the dysexecutive group significantly differed from the control group, and this effect remained after controlling for demographic variables, self-esteem, depression, and anxiety.

Two studies utilized the Measure of Insight into Cognition (MIC-SR) while assessing the accuracy of self-reported cognition. Medalia, Thysen, and Freilich [15] investigated self-reported cognitive insight in a sample that included only participants with SSDs who exhibited cognitive impairment. The researchers found that the MIC-SR did not correlate with the BACS or the Problem Solving subscale of the Independent Living Scales (ILS). Saperstein, Thysen, and Medalia [19] also found poor correspondence between the MIC-SR and the WAIS (Weschler Adult Intelligence Scale) Working Memory Index, though they found good convergence of the self-reported and clinician-rated versions of the MIC. When the sample was dichotomized based on clinician-rated MIC awareness, a group with good awareness of functioning reported significantly more cognitive difficulties than the group with poor awareness.

Six studies compared various self-report methods to neuropsychological assessments. Zanello and Huguelet [14] compared a neuropsychological battery and the Frankfurt Complaint Questionnaire (FCQ2). They correlated scores for the FCQ2 and all neuropsychological tests and found only one significant relationship between the perception and motor disorders factor of the FCQ2 and d2 percentage of errors, which emerged only after controlling for age. Donohoe et al. [16] divided participants into an “intact” clinical insight subgroup and an “impaired” clinical insight subgroup based on a semi-structured interview that assesses awareness of illness. The researchers then correlated scores from the Cognitive Failures Questionnaire (CFQ) and a cognitive battery. For the impaired clinical insight group, scores on the CFQ did not significantly correlate with any neuropsychological measure, while for the intact clinical insight group, CFQ scores correlated with measures of episodic memory and general cognitive functioning but not working memory or attentional control. During a validation study to evaluate the psychometric properties of the Subjective Cognitive Impairment Scale (SCIS), Chang et al. [21] correlated the SCIS with a neuropsychological assessment battery. Correlations largely failed to reach significance except for moderate relationships between the SCIS and both the letter-number span and PANSS cognition.

Tercero et al. [23] explored the concordance of self-reported cognitive ability using a modified version of the WCST, where participants were asked to indicate whether they believed they answered correctly and to rate how confident they were in their self-judgments. They found that participants with SSDs judged their performance as significantly higher than their actual performance and did not seem to incorporate feedback about their performance into global self-estimates of accuracy. Instead, global estimates of accuracy appeared to be related to trial-by-trial self-estimates of accuracy. Haugen and colleagues [24] created discrepancy scores by comparing the self-reported Behavior Rating Inventory of Executive Function (BRIEF-A) to performance on a cognitive battery and examined rates of under- and overestimation. They found that 48.5% of their sample overestimated their cognitive functioning, while 39.4% underestimated their functioning. Lastly, Morgan and colleagues [25] calculated introspective accuracy (IA) scores by asking participants to evaluate their performance directly after completing subtests of a cognitive battery. They found that absolute IA scores were significantly different from 0. However, subjective estimates correlated with actual performance for three of the four subtests.

Self-reported cognition appears to often be discordant with neuropsychological tests [15,17,19,20], though greater cognitive insight may increase correspondence [16,19]. Importantly, one study found that SSD groups with better executive functioning estimated their cognitive ability similarly to a healthy control group [22], and another found good concordance between subjective estimates and performance [25].

### 3.2. Self-Reported Everyday Functioning/Functional Capacity

Eight papers used the Subjective Level of Functioning Scale (SLOF) to assess the accuracy of self-reported everyday functioning or functional capacity. Bowie and colleagues [26] examined the SLOF scores of participants and caregivers, finding that correlations between self- and informant-reported scores on the SLOF were generally modest; only the personal care SLOF scales yielded correlations above *r* = |0.30|. When looking at discrepancies between self and informant ratings, these researchers found that 36% of their sample were relatively accurate in their self-estimations (i.e., rated themselves similarly to informants), 40% overestimated their functioning, and 24% underestimated their functioning. Sabbag et al. [27] used either high-contact clinicians or friends or relatives as informants. Both participants and informants completed the SLOF, the Social-Behavior Schedule (SBS), the Social Functioning Scale (SFS), the Life Skills Profile (LSP), and the ILSS. Overall, the researchers found mean differences between self-reports and reports by friends and relatives for the SBS and SLOF but not for other scales. There were no mean differences between patient reports and clinician reports. In another investigation, Sabbag et al. [28] found significant differences between self- and informant-reported scores for two of three SLOF subscales: vocational functioning and community activities. Self- and informant reports were comparable on the interpersonal relations subscale, and overall, 40% of the participants reported identical SLOF scores to their respective informants. Gould et al. [29] split their sample based on whether they had achieved different functional milestones and examined self-reported data from SLOF subscales that corresponded to the various milestones. When comparing SLOF interpersonal relations to marital status and SLOF work skills to past employment, no differences were found between groups who had and had not achieved these milestones. However, they found significant differences between groups who were currently employed or not on the SLOF work skills subscale, as well as differences on the SLOF everyday activities subscale when comparing those who were living independently or not and those who were financially responsible for their house or not. Harvey et al. [31] compared self-reported functioning to clinician-reported functioning. Across all three domains measured by the SLOF, participants reported higher levels of functioning on average compared to clinician ratings.

In a sixth SLOF study, Harvey and researchers [33] used self- and informant- or researcher-reported SLOF scores to derive introspective accuracy scores or scores that capture the magnitude of discordance between subjective and objective reports, and introspective bias scores, or scores that indicate both the magnitude of discordance as well as whether participants overestimated or underestimated their functioning. The researchers found that introspective accuracy scores were statistically different from 0 for all SLOF scales but that the introspective bias scores were symmetrically distributed around 0, indicating that participants both over- and under-estimated performance. Durand and colleagues [34] collected self-reported and either informant-reported or researcher-rated scores on the SLOF interpersonal functioning subscale. They found that participants with SSDs overestimated their social functioning compared to informants and researchers; they further reported that, when compared to a subgroup with bipolar disorders, individuals with SSDs did not overestimate their functioning more than those with bipolar disorders. Finally, Rocca and colleagues [35] also used self- and informant-reported SLOF scores to investigate concordance. When comparing subjective and objective reports using *t*-tests, they found small but statistically significant differences between sources such that participants with SSDs tended to overestimate their functioning. However, when looking at Bland–Altman limits of agreement, they found that about five-sixths of the individuals with SSDs rated themselves in concordance with informants, and the remaining portion of their sample tended to be poorly concordant within a single domain of functioning.

Four other papers looked at functional capacity using other assessments. Harvey et al. [30] compared a subset of questions from the ILSS related to financial management to two functional capacity assessments with financial performance subscales, the UPSA-B and the Everyday Functioning Battery (EFB). Performance on tests of functional capacity were not reflective of differences between self-reported financial management; those who reported having performed various financial tasks did not perform better than those who had not. Olsson et al. [32] used performance on the UPSA-B in conjunction with self-rated ability to perform several functional tasks to sort participants into four categories: accurate estimators with unimpaired functioning, underestimators with unimpaired functioning, overestimators with impaired functioning, and accurate estimators with impaired functioning. Fifty-six percent of the sample were found to be accurate estimators of their functioning (24% with unimpaired functioning and 32% impaired), 38% overestimated their functioning, and 6% underestimated their functioning. To evaluate social functioning, Jongs and colleagues [36] compared self- and informant or researcher-reported ratings on a subset of questions from the WHO Disability Assessment Schedule. They found that participants with SSDs reported significantly higher social functioning scores than caregivers and researchers. Okada [37] also looked at social functioning by dividing participants into over-estimation, accurate estimation, and under-estimation groups based on the degree of discrepancy between the SFS and the Life Assessment Scale for Mental Illness. In this study, 31% of participants over-estimated their social functioning, 41% were concordant with informants in their self-estimations, and 28% under-estimated their functioning.

It is notable that, compared to papers examining cognitive ability, more papers that examined everyday functioning found greater concordance between subjective and objective measures [27,32,35,37]. More concrete skills, such as personal hygiene [26] or employment and daily activities among those currently employed or living independently [29], tended to be estimated more accurately than more abstract skills like interpersonal functioning. Although not all studies explicitly examined factors such as the current relevance of functional domains or quality of informant relationships, those that did found that correspondence was greater when the measured domain was relevant to the participant [29] or when informants were in high contact with participants [35].

### 3.3. Both Self-Reported Cognition and Everyday Functioning

Two papers total examined both the accuracy of cognition and of everyday functioning or functional capacity in conjunction. Durand et al. [38] examined self- and clinician-reported scores on the Cognitive Assessment Interview (CAI) and SLOF; correlations between self and informant scores were small, and a series of *t*-tests showed significant differences between self- and clinician-reports on all measures. Gould et al. [2] also used self-reported and clinician-reported versions of the SLOF and the CAI. The researchers found that most participants tended to overestimate their abilities compared to clinician reports, but a significant proportion rated themselves as performing at a level equivalent to or lower than clinician ratings (33% of the sample were accurately or underestimating their ability for SLOF work skills, 47% for SLOF everyday activities, 44% for SLOF social functioning, and 40% on the CAI).

### 3.4. Predictors of Self-Report Concordance

*Depression.* Four studies directly examined the impact of depressed mood on self-report concordance and found a consistent pattern. In general, depressed mood correlated with subjective ratings that were either more concordant with objective ratings or that tended towards underestimation of ability. When Bowie et al. [26] compared groups based on the accuracy of estimation, those who underestimated their performance reported higher scores on the Beck Depression Inventory (BDI) than either accurate estimators or overestimators; similarly, Sabbag and colleagues [28] found that BDI scores were negatively correlated with accuracy scores for SLOF vocational skills, indicating that participants who underestimated their functional performance relative to informants reported more symptoms of depression. Two studies [31,33] divided their sample into groups based on BDI scores to explore the impact of depression on self-assessment of functioning, and both found that individuals with moderate or high levels of depression were more likely to rate their functioning consistently with informants or underestimate their functioning compared to individuals with low levels of depressive symptoms.

A further subset of five studies considered the effect depressed mood had on self-assessment scores alone without considering concordance. Lecardeur et al. [17] found a modest correlation between depression and SSTICS scores, and Saperstein et al. [19] found that the depression/anxiety factor of the BPRS correlated with the self-reported MIC. After controlling for cognition, duration of illness, medication level, positive and negative symptoms, insight, and hopelessness, Sellwood et al. [20] found that dysphoria, as measured by the Hospital Anxiety and Depression Scale, significantly predicted SSTICS scores. Chang et al. [21] found a moderate correlation between the SCIS total score and the PANSS depression and anxiety score. Finally, both Sabbag et al. [28] and Durand [38] found correlations between BDI scores and scores on all three SLOF subscales. In each of these studies, authors found that higher levels of depression were related to lower self-assessments of cognitive and functional abilities.

Of all 12 papers in this review that examined the relationship between self-assessment and depression, only 3 [22,24,35] failed to find any significant relationship.

*Cognition*. Of the studies that investigated the impact of cognitive ability on the accuracy of self-assessment, most found that better cognition was associated with more concordant estimation or underestimation of cognitive and everyday functioning. Bowie et al. [26] found that underestimators had better scores on a cognitive battery than overestimators. Sabbag et al. [28] found that performance on the MCCB correlated with SLOF discrepancy scores for the vocational subscale and total score, with negative correlations indicating that better performance on the MCCB was associated with accurate or overestimation of functional ability; Gould et al. [2] found poorer cognition to be associated with greater misestimation on all SLOF subscales. Durand et al. [38] found lower cognition to be correlated with overestimation of functioning for work and everyday activities as assessed by the SLOF, and lower scores on the MCCB were correlated with greater overestimation of cognitive performance.

## 4. Discussion

This review examined published papers that assessed the accuracy of self-reported data by individuals with schizophrenia-spectrum disorders (SSD) in the domains of cognition and everyday functioning. Most papers found minimal to modest associations between self-reported cognition and everyday functioning and informant-reported or behaviorally assessed measures of the same domains. Our review is consistent with and expands upon two previously published reviews that each aggregate part of this literature [40,41]. Results indicate that self-reported functioning may better align with objective measures of functioning in domains that are relatively more concrete and directly observable (e.g., self-care or work skills) versus those that are more abstract (e.g., cognition or interpersonal functioning). Further, individuals with SSD appear to self-report higher levels of functioning compared to objective measures—in studies that examined both over- and under-estimation, individuals who underestimated their functioning were a minority compared to those who accurately estimated or overestimated their functioning. Depression emerged as a clear predictor of lower self-reported functioning, and cognitive ability appeared to improve concordance of self-report measures with objective measures.

A key question that many of the papers reviewed here seek to answer is whether self-reported information from individuals with SSD is reliable enough to be used in clinical and research settings. Authors of previous overviews have provided suggestions for increasing the concordance of self-reported information with objective measures by improving patients’ insight into symptoms [40,41]. We argue that lack of concordance among various clinical data sources is not inherently problematic or necessarily indicative of a need for individuals with SSD to change their perspectives on their functioning. In the broader clinical assessment literature, discrepancies among data sources are a common and expected methodological issue. Among other populations, such discrepancies are not viewed as simple measurement errors or a product of inaccurate self-assessment; instead, it is expected that different data sources will capture different dimensions of functioning and provide unique, meaningful information [42]. Systematic frameworks for integrating multiple information sources have been developed for use in other areas of clinical assessment [42,43,44], and future research is necessary to adapt existing frameworks for use with individuals with SSD. At present, the central clinical challenge is to understand and meaningfully utilize discrepant sources of clinical data, including PROs. In the interest of advancing recovery-oriented frameworks for using and eliciting subjective information from individuals with SSD, we offer the following recommendations.

### 4.1. Understanding Discrepancies between Subjective and Objective Data Sources

Practitioners and researchers should not expect subjective reports to fully align with the results of neuropsychological assessments or informant reports and should consider the potential for self-report data to capture aspects of functioning that are not apparent in testing situations or to clinicians and other informants. For instance, it is plausible that an individual’s social functioning is more fluid or skilled within the context of their everyday routines and social networks, yielding a higher self-reported level of social functioning in comparison to clinician ratings, which may be based on observations within a clinical context that differs from everyday life. The preponderance of overestimation of everyday functioning in the reports within this review may, in part, reflect this. Further, practitioners and researchers should keep in mind that objective measures of functioning have their own limitations and can be susceptible to bias [45,46]. The results of neuropsychological or performance-based tests are influenced by various contextual factors such as lack of motivation, concerns about the outcome of the assessment, momentary influences like feeling distracted, as well as social influences like incongruence between the race of the assessor and person being assessed [47,48]. These factors would be expected to be differentially related or unrelated to an individual’s cognitive and functional abilities across time and circumstances, complicating the interpretation of results; this is perhaps one reason why past reviews have found only moderate associations between neuropsychological assessments and real-world outcomes or functioning [45]. In this review, self-reported cognition was frequently discordant with neuropsychological tests [15,17,19,20], but no reports measured momentary influences that may further explain the discrepancy. Future studies may benefit from including this information in their analyses.

Clinician and informant ratings of functioning can also be problematic. While there is the potential benefit of informants being able to observe an individual across settings or apply clinical judgment, clinician and informant ratings are not always highly correlated with neuropsychological measures [27,46,49], and informant-reported information carries the additional concern that patient characteristics may color impressions. One study in this paper raises the possibility that confident individuals who overestimate their functioning may be rated higher in functional domains by others as well, potentially resulting in them not receiving necessary support [32].

Unlike other data sources, an individual reporting on their own life has access to information across a wide range of situations and contexts, which may enhance the utility of self-reported information to predict real-world outcomes. For example, individuals may have found effective coping tools to overcome cognitive and functional difficulties in daily life that are not available in the testing environment (e.g., writing lists to cope with poor working memory or using online services to schedule medical appointments if using a telephone is challenging). A laboratory-based measure may then suggest greater impairment, even though they are able to successfully perform tasks when compensatory strategies are available to them in everyday life. Self-reports predict important treatment outcomes, like future psychiatric hospitalizations, imprisonment, incidents of self-harm, and reliance on social services [11,50]. While not included in typical assessments of functioning, remaining out of inpatient care and using fewer social services are indicators of functional ability. Numerous studies also find that PROs are predictive of treatment adherence, engagement, and satisfaction [51,52]. Research concerning other populations has highlighted the potential for self-reported data to detect future functional difficulties, as subjective judgments that underestimate performance may be an early indicator that greater cognitive effort is being used to achieve similar functional results [53,54].

### 4.2. Individual Factors to Consider When Interpreting Subjective Data

When using self-reported information about cognitive and functional abilities, several variables should be considered. In this review, depression stood out as an important factor in self-report accuracy. Six of the papers in this review found that higher levels of depressed mood or sadness corresponded with lower self-estimates of functioning. This is congruent with research in other populations, which has shown that individuals with more depressive symptoms tend to underestimate subjective cognition [55,56]. Further, Kinsinger et al. [57] demonstrated that a decrease in depression symptoms improved the overall accuracy of subjective cognition. Though we cannot be sure if the influence of depression truly increases insight into symptoms, depressive symptoms attenuate subjective estimates of cognition and functional ability and should be considered when using self-reported data to interpret study results or guide treatment decisions.

It is important to consider an individual’s clinical history when evaluating self-reported data. As highlighted in Rocca et al. [35], symptom stability and consistent engagement in care are likely to promote better awareness of one’s functional abilities and limitations, possibly as a function of metacognitive ability [58]. Researchers and clinicians should keep in mind that challenges with self-reported information among the general population may intersect with difficulties faced by clinical populations. For example, studies have demonstrated that demographic factors like age and gender can impact estimations of cognitive performance [59,60]. Other factors relevant to clinical populations may influence an individual’s ability to report on their functioning but remain understudied, such as cognitive insight [61], metacognitive ability [62], general negative symptoms [36], disorganized symptoms [24], and symptoms of autism [33]. Insight stood out as a potential moderator in this review when considering self-reported cognition [16,19].

### 4.3. Recommendations for Eliciting Subjective Information

Clinicians and future studies that seek to construct, validate, and utilize self-reported measures for individuals with schizophrenia should carefully consider how they elicit information. Though research into psychometrically sound PROs is growing [63], some methodological features of current self-report inventories may limit their utility. One review of self-reported quality-of-life inventories among individuals with psychosis found that many measures were vague and had little evidence of psychometric quality [64]. Research concerning accessible assessment of PROs among populations that may struggle with attention, memory, or executive dysfunction suggests adapting design features to reduce assessment burden, such as giving specific examples or extra direction during administration, using visual aids where appropriate, or reducing cognitive demands by laying out measures in ways that reduce the need to visually scan and shift attention during administration [65,66]. Especially in populations where cognitive difficulties are common, care should be taken to ensure that participants adequately understand questionnaire items and that the layout, instructions, and response options of the measure are not overly burdensome. Assisting participants by providing extra verbal instructions as needed may be appropriate.

Self-report accuracy may also be facilitated by assessing functioning in ways that are both specific and temporally proximal to when the skill or behavior is practiced. Self-report inventories often contain questions that measure multiple domains of functional or cognitive abilities, which may result in estimates of functioning that are vaguer than what is elicited by performance-based assessment of a task; this may contribute to a lack of concordance with functional measures [67]. In this review, several papers found that self-report accuracy was improved when asking about behaviors that were concrete and relevant to participants, such as personal hygiene [26] or work skills among individuals currently employed [2]. Similarly, other studies have found that the accuracy of self-reported ability is improved when individuals are asked to estimate ability at the time of skill performance [68]. We therefore recommend closely examining self-report measures to ensure that items ask about specific behaviors before administration and considering when individuals may have last performed a skill when interpreting the results. The best practice may be to solicit estimates of ability directly after skill performance when possible.

### 4.4. Strengths and Limitations

This paper provides a comprehensive review of studies that explore discrepancies between subjective and objective reports of cognitive and functional ability among individuals with SSD. We expand on previous reviews by compiling quantitative results from the included studies and providing specific recommendations for working with subjective reports from this population. We emphasize that while caution should be exercised regarding the accuracy of self-assessment of cognition and functioning among some individuals with schizophrenia based on the available data, equal caution should be exercised regarding the accuracy of measures commonly considered to be objective or true measures of cognition and functioning. We suggest methods for increasing the validity of objective assessments and point to the need to consider how contextual factors may affect an individual’s ability to reflect on their functioning. Additionally, we provide a recovery-oriented perspective regarding self-assessment among individuals with SSD, as self-reports predict important outcomes regardless of their alignment with objective measures.

This review had several limitations. Because of heterogeneity across studies, we were unable to perform a quantitative meta-analysis that may better identify domains and predictors of self-report accuracy. Further, the risk of bias was a concern for many of the reports included in this review. Our assessment revealed that the included studies often failed to report key information: only 27% discussed participant exclusion, 19% reported the validity of measures used, 15% reported any kind of data management strategies, and no papers adequately described their participant recruitment methods. We have included relevant statistics for each study to aid future researchers in evaluating these papers and our interpretations. Another possible source of bias is the exclusion of unpublished reports in this review. Trial registers are not frequently available for reports that are not clinical trials, as many studies addressing this topic are, which limits the ability to identify unpublished work. This may result in skewed estimates of the correspondence between subjective and objective data reported in this review. The reports included also originated primarily from the USA, which may influence the results presented. There is also an overall dearth of information on the specific question addressed by this review. As recovery-oriented services for people with schizophrenia become more available and self-report inventories more frequently used in research and practice, best practices for using self-reported data will continue to be an important research area.

## 5. Conclusions

Our review does not contradict prior concerns about the limited concordance between self-reported and objectively assessed cognition and functioning. Indeed, our synthesis of the literature reveals that existing research has identified cognitive and clinical features that influence self-assessment. However, our review also underscores the tendency for research and clinical efforts to emphasize disease-related factors and be relatively inattentive to the contextual and assessment-related considerations that may impact discrepancies between various data sources. We argue that there is a need to move beyond frameworks that focus on remediating the self-assessment abilities of individuals with schizophrenia. Rather, comprehensive assessment and clinical frameworks are needed to integrate data among critical and often discrepant data sources. Such considerations can not only provide a more complete picture of factors related to self-report accuracy among some individuals with schizophrenia but also push research and clinical practices forward in ways that respect and honor client experiences of self.

## Figures and Tables

**Table 1 behavsci-14-00030-t001:** Results of Studies Comparing Subjective and Objective Cognition.

Authors	Study Design, Country	Sample		Measures			Results
				Subjective	Objective	Predictors	
Zanello and Huguelet, 2001 [14]	Cross-sectional, Switzerland	N ^‡^	50	FCQ2	Verbal Fluency, Stroop, d2		- Correlations among the FCQ2 and all included neuropsychological tests found only one significant relationship between the d2 percentage of errors and the perception and motor disorders factor of the FCQ2 (*r* = −0.31, *p* = 0.037), which emerged only after controlling for age.
		Age (M (SD))	30.7 (6.4)				
		Gender (% Male)	45.71				
Medalia et al., 2008 [15]	Cross-sectional, USA	N ^†^	71	MIC-SR	BACS, ILSS-PS		- The sample included only participants who exhibited cognitive impairment as determined by a priori cutoff scores on the BACS or WRAT; - Found that the MIC-SR did not correlate with the BACS (*r* = 0.06, *p* = 0.61) or the ILS-PS (*r* = 0.01, *p* = 0.93).
		Age (M (SD))	38.9 (11.4)				
		Gender (% Male)	73.24				
Donohoe et al., 2009 [16]	Cross-sectional, Ireland	N ^†^	51	CFQ	WAIS-III, WMS-III (LM, Faces, LNS), Cantab (Spatial Working Memory, IDED, Attention to Response)		- Participants were divided into an “intact” clinical insight subgroup and an “impaired” clinical insight subgroup based on a semi-structured interview, then scores from the CFQ and a cognitive battery were correlated; - For the impaired clinical insight group (*n* = 27), scores on the CFQ did not significantly correlate with any cognitive measures; - For the intact clinical insight group (*n* = 24), CFQ scores correlated with measures of episodic memory and general cognitive functioning (*r* = |0.37| − |.48|, *p* < 0.05) but not working memory (*r* = |0.12| − |0.22|) or attentional control (*r* = |0.03| − |0.29|).
		Age (M (SD))	NR				
		Gender (% Male)	NR				
Lecardeur et al., 2009 [17]	Cross-sectional, Canada	N ^§^	176	SSTICS, FPSES	PANSS Cognition	PANSS	- Found good convergence of the SSTICS with the FPSES (*r* = 0.54, *p* < 0.01); - The FPSES total score failed to correlate with any of the PANSS factors; - The SSTICS total score correlated with PANSS cognition (*r* = 0.34, *p* = 0.004); - Found a correlation between PANSS depression and SSTICS total (*r* = 0.26, *p* = 0.029).
		Age (M (SD))	33.5 (11.6)				
		Gender (% Male)	64.77				
Johnson et al., 2011 [18]	Cross-sectional, Tunisia	N ^‡^	104	SSTICS_tun_arab	Tunisian Cognitive Battery		- No significant correlations between the SSTICS total score and any of the neuropsychological tests (*r* = |0.01| − |0.12|) or a PANSS item assessing insight into cognitive symptoms (*r* = −0.14).
		Age (M (SD))	34 (7)				
		Gender (% Male)	81.73				
Saperstein et al., 2012 [19]	Cross-sectional, USA	N ^‡^	73	MIC-SR	MIC-CR	BPRS	- Poor correspondence was found between the MIC-SR and WMI; - Found good convergence of the self-reported and clinician-rated versions of the MIC (*r* = −0.70, *p* < 0.001); - A subgroup with good clinician-rated MIC awareness of functioning reported significantly more cognitive difficulties than the group with poor awareness (*t*(71) = 4.65, *p* < 0.001, d = 1.14); - Found that the depression/anxiety factor of the BPRS correlated with the self-reported MIC (ẞ = 0.38, *p* = 0.001).
		Age (M (SD))	39.40 (12.21)				
		Gender (% Male)	65.75				
Sellwood et al., 2013 [20]	Cross-sectional, United Kingdom	N ^§^	115	SSTICS	BACS	HADS	- No significant correlations between any SSTICS subscales and the BACS domain scores (*r* = |0.010| − |0.157|) or between the total SSTICS score and single factor BACS score (*r* = 0.006); - Only dysphoria, as measured by the HADS significantly predicted SSTICS scores (ẞ = 0.68, *p* < 0.001).
		Age (M (SD))	36.0 (11.59)				
		Gender (% Male)	73.04				
Chang et al., 2015 [21]	Cross-sectional, China	N ^§^	101	SCIS	PANSS Cognition, WAIS-R (DS), WMS-R (LM), TMT, MCST	PANSS	- SCIS self-report largely failed to correlate with NP assessments. - Correlated with PANSS cognition (*r* = 0.28, *p* = 0.005) and LNS (*r* = −0.33, *p* = 0.001); - PANSS depression and anxiety were moderately correlated with the SCIS (*r* = 0.41, *p* = 0.001).
		Age (M (SD))	25.0 (7.5)				
		Gender (% Male)	45.54				
Prouteau et al., 2015 [22]	Cross-sectional, France	N ^‡^	40	SSTICS	MCST		- Calculated discrepancy scores between SSTICS and MCST to represent neurocognitive insight among three participant groups: an “executively normal” (*n* = 29) subgroup, a “dysexecutive” subgroup (*n* = 11), and a control group without any psychiatric diagnosis (*n* = 42); - 63% of the dysexecutive group exhibited high overestimation of cognitive functioning; only 24% of the executively normal subgroup and 9.5% of the control group did; - The two schizophrenia subgroups (dysexecutive and executively normal) did not differ from each other in NI scores (*p* = 0.064, d = 0.63), nor did the executively normal group differ from the control group (*p* = 0.42, d = 0.33); - The dysexecutive group significantly differed from the control group (*p* < 0.05, d = 1.31).
		Age (M (SD))	37.77 (9.58)				
		Gender (% Male)	55				
Tercero et al., 2021 [23] ¶	Cross sectional, USA	N ^‡^	99	MCST	WCST		- Accuracy judgments were significantly higher for both schizophrenia and bipolar disorder participants than actual WCST performance (*t* > 9.57; *p* < 0.001), reflecting significant impairment in introspective accuracy and positive introspective bias; - In a regression analysis where accuracy judgments predicted trial-by-trial confidence, the overall model for participants with schizophrenia was significant (*F*(1,97) = 48.49, *p* < 0.001), and accuracy judgments significantly predicted 33% of the variance; - In a second regression analysis where trial-by-trial confidence and accuracy judgments predicted global performance judgments, the overall model was again significant (*F*(2,94) = 38.95, *p* < 0.001) with trial-by-trial confidence accounting for 26% of the variance and accuracy judgments accounting for 20%; - WCST performance failed to enter either regression model, and the shared variance for performance and accuracy judgments was 4% among participants with SCZ.
		Age (M (SD))	41.98 (10.44)				
		Gender (% Male)	52.53				
Haugen et al., 2021 [24]	Cross-sectional, Norway	N ^§^	66	BRIEF-A	Color-Word 3, D-KEFS, CPT3, DS, LNS	GPSES, PANSS	- In a comparison between BRIEF-A and a cognitive battery, 48.5% ranked lower on total subjective complaints than objective measures (stoicism or overestimation of ability), and 39.4% ranked higher on total subjective complaints than objective measures (sensitivity or underestimation of ability); - Higher levels of depressive symptoms did not predict greater sensitivity.
		Age (M (SD))	25.53 (6.56)				
		Gender (% Male)	60.61				
Morgan et al., 2022 [25]	Cross-sectional, USA	N ^†^	126	Accuracy Probe Questions	MCCB (TMT-A, LNS, ANT, HVLT)	Momentary positive symptoms measured via EMA	- Derived an introspective accuracy (IA) score by asking participants to estimate their performance after each subtest of the MCCB. The absolute IA score (M = 0.85, SD = 0.37) was significantly greater than 0 for participants with schizophrenia (*t*(125) = 12.98, *p* < 0.001, d = 1.10). - Self-reported performance was significantly correlated with actual performance on the TMT-A (*r* = 0.35, *p* < 0.001), ANT (*r* = 0.45, *p* < 0.001), and HVLT (*r* = 0.47 *p* < 0.001) but not LNS (*r* = 0.13, *p* = 0.17). - IA was negatively correlated with performance on the LNS (*r* = −0.51, *p* < 0.001), ANT (*r* = −0.27, *p* = 0.002), and HVLT (*r* = −0.25, *p* = 0.005) but not the TMT-A (*r* = −0.13, *p* = 0.17).
		Age (M (SD))	41.90 (10.74)				
		Gender (% Male)	51.59				

Note: † denotes sample diagnosed with schizophrenia only, ‡ denotes sample diagnosed with schizophrenia or schizoaffective disorders, § denotes sample diagnosed with various schizophrenia-spectrum disorders. ¶ denotes that this publication appears to be a paper from the VALERO I or II studies. ANT = Animal Naming Test, BACS = Brief Assessment into Cognition in Schizophrenia, BPRS = Brief Psychiatric Rating Scale, BRIEF-A = Behavior Rating Inventory of Executive Function—Adult, CFQ = Cognitive Failures Questionnaire, CPT3 = Continuous Performance Test, d2 = d2 Test of Cognition, D-KEFS = Delis–Kaplan Executive Function System, DS = Digit Span, FCQ2 = Frankfurt Complaint Questionnaire-2, FPSES = Frankfurt-Pamplona Subjective Experiences Scale, GPSES = General Perceived Self-Efficacy Scale, HADS = Hospital Anxiety and Depression Scale, HVLT = Hopkins Verbal Learning Test, ILSS-PS = Independent Living Scale—Problem Solving, LM = Logical Memory, LNS = Letter-Number Span, MCCB = MATRICS Consensus Cognitive Battery, MCST = Modified Card Sorting Task, MIC-SR/CR = Measure of Insight into Cognition—Self-Report/Clinician-Report, PANSS = Positive and Negative Syndrome Scale, SCIS = Subjective Cognitive Impairment Scale, SSTICS = Subjective Scale to Investigate Cognition in Schizophrenia, SSTICS_tun_arab = Subjective Scale to Investigate Cognition in Schizophrenia Tunisian Arabic Version, TMT = Trail Making Test, WAIS-III/R = Wechsler Adult Intelligence Scale—Third Edition/Revised, WCST = Wisconsin Card Sorting Task, WMI = Working Memory Index, WMS-III/R = Weschler Memory Scale—Third Edition/Revised, WRAT-3 = Wide Range Achievement Test-3rd Edition.

**Table 2 behavsci-14-00030-t002:** Results of Studies Comparing Subjective and Objective Everyday Functioning and Functional Capacity.

Authors	Study Design, Country	Sample		Measures			Results
				Subjective	Objective	Predictors	
Bowie et al., 2007 [26]	Cross-sectional, USA	N ^‡^	67	SLOF	SLOF	BDI, TMT, WAIS-III (Digit Symbol Coding, Digit Span, LNS) RAVLT, COWAT, Stroop, WRAT	- Correlations between self- and informant-reported scores on the SLOF were generally modest; only personal care, as reported by case managers, yielded correlations with self-reported SLOF scales above *r* = |0.30|; - 36% of the sample were relatively accurate in their self-estimations (i.e., rated themselves similarly to informants), 40% overestimated their functioning, and 24% underestimated their functioning; - Underestimators reported higher BDI scores than either accurate estimators or overestimators (*F*(2,66) = 4.8, *p* = 0.01); - Underestimators had better scores on a cognitive battery than overestimators (*F*(2,66) = 3.3, *p* = 0.04).
		Age (M (SD))	56.6 (7.5)				
		Gender (% Male)	76.12				
Sabbag et al., 2011 [27] ¶	Cross-sectional, USA	N ^‡^	193	SLOF, ILSS, QLS, SBS, SFS, LSP	SLOF, ILSS, QLS, SBS, SFS, LSP		- Found mean differences between self-reports and reports by friends and relatives for the SBS (*t*(153) = −3.63, *p* = 0.000, d = 0.43) and SLOF (*t*(153) = 2.808, *p* = 0.006, d = 0.35), but not for other scales; - No mean differences between patient reports and clinician reports; - Self-reports and friend or relative reports were correlated for the SBS (*r* = 0.25, *p* = 0.007), the SFS (*r* = 0.48, *p* = 0.000), and marginally for the ILSS (*r* = 0.19, *p* = 0.05). Only the correlation between patient and clinician scores for the SFS reached significance (*r* = 0.41, *p* = 0.03).
		Age (M (SD))	44.08 (11.69)				
		Gender (% Male)	68.92				
Sabbag et al., 2012 [28] ¶	Cross-sectional, USA	N ^‡^	121	SLOF	SLOF	MCCB, BDI, UPSA-B	- Found significant differences between self- and informant-reported scores for two of three SLOF subscales: vocational functioning (*t* = 6.48, *p* = 0.001, d = 0.46) and community activities (*t* = 5.29, *p* = 0.001, d = 0.37); - Self- and informant reports were comparable on the interpersonal relations subscale (*t* = 1.43, *p* = 0.17, d = 0.18); - 40% of the participants reported identical SLOF scores to their respective informants; - Performance on the MCCB correlated with SLOF discrepancy scores for the vocational subscale (*r* = −0.21 *p* < 0.01) and the total score (*r* = −0.17 *p* < 0.05), indicating that better performance on the MCCB was associated with accurate or overestimation of functional ability; - BDI scores were negatively correlated with discrepancy scores for SLOF vocational skills (*r* = −0.16, *p* < 0.05); - Higher levels of self-reported depression were associated with lower scores on self-reported SLOF (*r* = −0.31 to -.51, *p* < 0.001); - UPSA-B correlated with discrepancy scores on the SLOF vocational subscale (*r* = −0.21, *p* < 0.01).
		Age (M (SD))	44.03 (11.73)				
		Gender (% Male)	NR				
Gould et al., 2013 [29] ¶	Cross-sectional, USA	N ^‡^	195	SLOF	Functional Milestones		- The sample was split based on whether they had achieved different functional milestones; - When comparing SLOF interpersonal relations to marital status and SLOF work skills to past employment, no differences were found between groups who had and had not achieved these milestones; - Significant differences were found between groups who were currently employed or not on the SLOF work skills subscale (*t* = 3.27, *p* = 0.001, d = 0.78); - Found significant differences between those who were living independently or not (*t* = 2.80, *p* = 0.006, d = 0.38) and those who were financially responsible for their house or not (*t* = 2.66, *p* = 0.008, d = 0.40) on the SLOF everyday activities subscale.
		Age (M (SD))	44.03 (11.73)				
		Gender (% Male)	69				
Harvey et al., 2013 [30] ¶	Cross-sectional, USA	N ^†^	195	ILSS	ILSS (Informant Form), UPSA-B (Finances), EFB (Advanced Finances)	MCCB	- Participants who reported having performed various financial tasks did not perform differently on tests of functional capacity (EFB, UPSA-B) than those who had not; - MCCB scores did not significantly differ between those who reported having performed financial tasks.
		Age (M (SD))	44.03 (11.73)				
		Gender (% Male)	69				
Harvey et al., 2017 [31] ¶	Cross-sectional, USA	N ^‡^	406	SLOF	SLOF	BDI	- Across all three domains measured by the SLOF, participants reported higher levels of functioning on average compared to clinician ratings (d = 0.33–0.67); - Split sample into three groups based on their scores on the BDI; found that individuals in the moderate depression group (*n* = 127) and high depression group (*n* = 129) were more likely to rate their functioning similarly to clinicians than those with low depression (*n* = 150) on all three SLOF subscales (interpersonal functioning: *F* = 5.10, *p* = 0.007; everyday activities: *F* = 6.34, *p* = 0.002; vocational skills: *F* = 6.31, *p* = 0.002).
		Age (M (SD))	42.3 (12.2)				
		Gender (% Male)	66.01				
Olsson et al., 2019 [32]	Cross-sectional, Sweden	N ^§^	222	Self-Rated Functioning (Prior to UPSA)	SLOF, UPSA-B		- Used the UPSA-B and self-rated ability to perform several functional tasks to sort participants into four categories: accurate estimators with unimpaired functioning (as measured by UPSA-B), underestimators with unimpaired functioning, overestimators with impaired functioning, and accurate estimators with impaired functioning; - 56% of the sample were found to be accurate estimators of their functioning (24% with unimpaired functioning and 32% impaired), 38% overestimated their functioning, and 6% underestimated their functioning; - Found that accurate estimators with unimpaired functioning had better executive functioning Wilks’ (λ = 0.657, *F*(2,60) = 15.66 *p* < 0.001 Wilks’ λ = 0.632, *F*(2,56) = 16.31 *p* < 0.001) than accurate estimators with impaired functioning and overestimators with impaired functioning.
		Age (M (SD))	51.67 (11.50)				
		Gender (% Male)	62.16				
Harvey et al., 2019 [33]	Cross-sectional, USA	N ^‡^	177	SLOF	SLOF	BDI, PANSS	- Found that introspective accuracy scores for all SLOF scales were statistically different from 0 (interpersonal functioning: *t* = 15.43, *p* < 0.001, d = 1.18; everyday activities: *t* = 10.25, *p* < 0.001, d = 0.77; vocational functioning: *t* = 13.86, *p* < 0.001, d = 1.16) but that the range of introspective bias scores was nearly symmetrical around 0, suggesting that participants were inaccurate but tended to both over- and under-estimate their performance; - When participants were split into low, medium, and high depression groups, differences in introspective bias were present between groups for all SLOF subscales (interpersonal functioning: *F*(3,175) = 4.06, *p* = 0.019; everyday activities: *F*(3,175) = 3.90, *p* = 0.022; vocational functioning: *F*(3,175) = 7.54, *p* = 0.001); - For everyday activities, low/moderate depression was associated with scores consistent with informant ratings; the most depressed group underestimated performance. For interpersonal functioning, low/moderate depression was associated with overestimates of functioning; the most depressed group was consistent with informant ratings. For vocational functioning, differences in all groups: low depression overestimated functioning, high depression underestimated functioning, and moderate was consistent with informant ratings.
		Age (M (SD))	40.5 (11.5)				
		Gender (% Male)	54				
Durand et al., 2021 [34]	Cross-sectional, USA	N ^‡^	102	SLOF interpersonal functioning	SLOF interpersonal functioning		- Participants with Schizophrenia significantly overestimated their functioning on SLOF interpersonal functioning (M = 1.67; SD = 5.87; *t*(101) = 2.45, *p* = 0.012, d = 0.28); - Participants with schizophrenia did not significantly overestimate their functioning compared to participants with BD (M_SCZ_ 1.67 > M_BD_ −0.60; *t*(172) = 1.85, *p* = 0.06, d = 0.40).
		Age (M (SD))	41.98 (10.44)				
		Gender (% Male)	51.96				
Rocca et al., 2021 [35]	Cross-sectional, Italy	N ^†^	618	SLOF	SLOF	PANSS, BNSS	- Found statistically significant differences between self- and informant-reported SLOF domain scores (interpersonal relationships: *t*(617) = 6.36, *p* < 0.001, Somers’ *D* = 0.25; everyday life skills: *t*(617) = 5.50, *p* < 0.001, Somers’ *D* = 0.27; work skills: *t*(617) = 6.00, *p* < 0.001, Somers’ *D* = 0.27). However, the magnitude of differences was small (0.83–1.04 points); - Good concordance between self-reported functioning and informant reports on SLOF as measured by Bland-Altman LOA: five out of six patients were in concordance with their caregivers. 17.6% of patients were poorly concordant; - The sample comprise of highly stable outpatients supported with good care, and informants were caregivers close to participants.
		Age (M (SD))	45.1 (10.5)				
		Gender (% Male)	69.09				
Jongs et al., 2022 [36]	Cross-sectional, Spain and the Netherlands	N ^†^	61	WHODAS social functioning	WHODAS social functioning	PANSS	- Schizophrenia patients reported their social functioning scores to be significantly higher than those reported by caregivers and researchers (Mann-Whitney *U* = 680.5, *n* = 54, *p* < 0.004).
		Age (M (SD))	30.13 (6.55)				
		Gender (% Male)	67.21				
Okada, 2022 [37]	Cross-sectional, Japan	N ^‡^	100	SFS	LASMI	BPRS, BNSS, SCoRS, SFS, Vocational Outcomes	- Divided participants into over-estimation, accurate estimation, and under-estimation groups based on the discrepancy between the SFS and the LASMI; 31% overestimated their social functioning, 41% were accurate, and 28% underestimated.
		Age (M (SD))	47.31 (12.93)				
		Gender (% Male)	NR				

Note: † denotes sample diagnosed with schizophrenia only, ‡ denotes sample diagnosed with schizophrenia or schizoaffective disorders, § denotes sample diagnosed with various schizophrenia-spectrum disorders. ¶ denotes that this publication appears to be a paper from the VALERO I or II studies. BCIS = Beck Cognitive Insight-Scale, BDI = Beck Depression Inventory, BPRS = Brief Psychiatric Rating Scale, COWAT = Controlled Oral Word Association Test, EFB = Everyday Functioning Battery, ILSS = Independent Living Skills Survey, LNS = Letter-Number Span, LSP = Life Skills Profile, MAS-A = Metacognition Assessment Scale-Abbreviated, MCCB = MATRICS Consensus Cognitive Battery, PANSS = Positive and Negative Syndrome Scale, QLS = Quality of Life Scale, RAVLT = Reyes Auditory Learning Test, SBS = Social-Behavior Schedule, SCSQ = Social Cognition Screening Questionnaire, SFS = Social Functioning Scale, SLOF = Specific Levels of Functioning Scale, TMT = Trail Making Test, UPSA = UCSD Performance-Bases Skills Assessment, WAIS-III = Wechsler Adult Intelligence Scale—Third Edition, WBI = Work Behavior Inventory, WRAT = Wide Range Achievement Test.

**Table 3 behavsci-14-00030-t003:** Results of Studies Comparing Subjective and Objective Cognition and Everyday Functioning and Functional Capacity.

Authors	Study Design, Country	Sample		Measures			Results
				Subjective	Objective	Predictors	
Durand et al., 2014 [38] ¶	Cross-sectional, USA	N ^‡^	207	SLOF, CAI	SLOF, CAI	BDI, MCCB	- A series of *t*-tests showed significant differences between self- and clinician reports on SLOF and CAI; patients tended to overestimate; - Correlations between self and informant scores on the CAI and SLOF yielded very small correlations (*r* = |0.01| − |0.12|; - Significant correlations were found between BDI scores and scores on all three self-reported SLOF subscales and the CAI (*r* = |0.26| − |0.44|); - Self-reported SLOF did not correlate with UPSA-B (*r* = |0.00-12|), and self-reported CAI did not correlate with MCCB (*r* = 0.12).
		Age (M (SD))	41.0 (12.4)				
		Gender (% Male)	67.63				
Gould et al., 2015 [2] ¶	Cross-sectional, USA	N ^‡^	214	SLOF, CAI	SLOF, CAI	MCCB	- Found that most participants tended to overestimate their abilities compared to clinician reports; - A significant proportion rated themselves as performing at a level equivalent to or lower than clinician ratings (33% of the sample were accurately or underestimating their ability for SLOF work skills, 47% for SLOF everyday activities, 44% for SLOF social functioning, and 40% on the CAI); - Poorer subjective cognition, as measured by the CAI, was associated with greater misestimation on all SLOF subscales (*r* = 0.39–0.49, *p* < 0.01); - Poorer performance on the MCCB was associated with greater misestimation on the SLOF interpersonal functioning (*r* = −0.20, *p* < 0.01) and everyday activities (*r* = −0.24, *p* < 0.01) subscales but not the vocational functioning subscale.
		Age (M (SD))	41.0 (12.4)				
		Gender (% Male)	64.95				

Note: ‡ denotes sample diagnosed with schizophrenia or schizoaffective disorders. ¶ denotes that this publication appears to be a paper from the VALERO I or II studies. BACS = Brief Assessment into Cognition in Schizophrenia, BCIS = Beck Cognitive Insight-Scale, BDI = Beck Depression Inventory, BDI-II = Beck Depression Inventory—Second Edition, BDI-III = Beck Depression Inventory—Third Edition, BHS = Beck Hopelessness Scale, BLERT = Bell-Lysaker Emotional Recognition Task, BPRS = Brief Psychiatric Rating Scale, CAI = Cognitive Assessment Inventory, CFQ = Cognitive Failures Questionnaire, CGI = Clinical Global Impression, COWAT = Controlled Oral Word Association Test, CPT-IP = Continuous Performance Test Identical Pairs, d2 = d2 Test of Cognition, EFB = Everyday Functioning Battery, FCQ2 = Frankfurt Complaint Questionnaire-2, FPSES = Frankfurt-Pamplona Subjective Experiences Scale, GAF = Global Assessment of Functioning, HADS = Hospital Anxiety and Depression Scale, ILSS = Independent Living Skills Survey, ILS-PS = Independent Living Scale—Problem Solving, IS = Birchwood Insight Scale, LNS = Letter-Number Span, LSP = Life Skills Profile, MAS-A = Metacognition Assessment Scale-Abbreviated, MCCB = MATRICS Consensus Cognitive Battery, MCST = Modified Card Sorting Task, MIC-SR/CR = Measure of Insight into Cognition—Self-Report/Clinician-Report, OPIE = Oklahoma Premorbid Intelligence Estimate, PANSS = Positive and Negative Syndrome Scale, PPFS = Patient Perception of Functioning Scale, QLS = Quality of Life Scale, QOL = Quality of Life Scale, RAVLT = Reyes Auditory Learning Test, RFS = Role Functioning Scale, RSE = Rosenberg Self-Esteem Inventory, SBS = Social-Behavior Schedule, SCIS = Subjective Cognitive Impairment Scale, SCSQ = Social Cognition Screening Questionnaire, SFS = Social Functioning Scale, SLOF = Specific Levels of Functioning Scale, SOFAS = Social and Occupational Functioning Assessment Scale, SSPA = Social Skills Performance Assessment, SSTICS = Subjective Scale to Investigate Cognition in Schizophrenia, SSTICS_tun_arab = Subjective Scale to Investigate Cognition in Schizophrenia Tunisian Arabic Version, STAI = State-Trait Anxiety Inventory, TMT = Trail Making Test, UPSA = UC*SD* Performance-Bases Skills Assessment, UPSA-B = UC*SD* Performance-Bases Skills Assessment—Brief, WAIS-III = Wechsler Adult Intelligence Scale—Third Edition, WAIS-R = Wechsler Adult Intelligence Scale—Revised, WCST = Wisconsin Card Sorting Task, WMI = Working Memory Index, WMS-III = Weschler Memory Scale—Third Edition, WMS-R = Weschler Memory Scale—Revised, WBI = Work Behavior Inventory, WRAT-3 = Wide Range Achievement Test, Third Edition.

## Data Availability

The data that support the findings of this study are available from the corresponding author upon reasonable request.

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
