# Peer review of "The Discrepancy between Subjective and Objective Evaluations of Cognitive and Functional Ability among People with Schizophrenia: A Systematic Review"

_behavsci, 2023, doi:10.3390/bs14010030_

Round 1
Reviewer 1 Report
Comments and Suggestions for Authors
The authors present a systematic review of the concordance among subjective and objective measures (performance- or informant-based) of cognition and daily functioning. The methods are well-detailed, the compilation of findings is informative, and the discussion pertaining to field recommendations is rich. I offer the following considerations that may help to further bolster the report:
1. The methods are well-described and appear to largely adhere to most reporting guidelines/ recommendations. The authors could more explicitly indicate whether they follow any particular set – e.g., to what extent to their methods adhere to the latest PRISMA methods (see http://www.prisma-statement.org/ and updates summarized by Page et al., 2021, BMJ, 372, n71. doi: 10.1136/bmj.n71).
2. Related questions relate to why the authors only focused on published articles and excluded ‘grey’ literature, and how they determined the key words used for their search strategy as this defines the scope.
3. While some effect sizes are provided, there are several results in the Tables where the authors could calculate effect sizes from the stats provided (e.g., t tests) or may be able to extract effect sizes from data in the studies (e.g., means, SDs).
4. The Results section reads as a very descriptive summary of the findings reported in the Tables. Either in the Results or in the Discussion, it could benefit the review to provide more synthesis. For example, after discussing the several studies in each section, a summary of the key patterns observed would help to provide the ‘take-home’ messages. The authors might also draw from the Tables/studies reviewed to offer observations from across the studies. For example, are there any notable trends for moderation of findings by performance vs. informant comparisons to subjective ratings, or by demographics (age, gender, country). It might also be noted that most studies appear to comprise a greater proportion of males, average middle-aged, conducted in the USA. This could also be reflected upon in the Limitations.
5. I found the Discussion to be particularly rich in providing valuable considerations and recommendations to the field for the interpretation of and approach to assessing subjective and objective functioning in SSD (although most of the arguments extend beyond SSD). At the same time, the bulk of this discussion seems largely independent of the review and its main objective. That is, other than having the basis of summarizing modest concordance rates, much of the Discussion seems like it could have been written separately. It largely presents the argument that discordance should be expected and that comparing subjective to objective measures in itself may not be the most meaningful exercise, yet it was the main research question guiding the review. Perhaps the point could be clarified in context. Also, while the review incorporates aspects of the findings in places (e.g., Section 4.2 notes depression and cognitive ability presented as noted moderators), but it could benefit from more integration of the findings to support the points being raised and recommended. Many of the recommendations, while much appreciated, extend beyond the ‘data’ summarized in the Results/Tables.
Author Response
- This review was originally designed following PRISMA guidelines prior to the 2020/2021 update. Effort has been made to include updated PRISMA recommendations where possible. A PRISMA statement had been added to the manuscript.
- As the review focuses on studies that are not clinical trials and therefore are not frequently registered, including "grey" literature was more difficult than might be expected for reviews of studies that are registered clinical trials. Efforts were made to include reports that may not have been identified in the original search, such as hand-searching the bibliographies of identified studies. This point has been added to the limitations section. Keywords were determined via collaboration among authors. This has been added to the methods section.
- The original review manuscript reported only effect sizes included in the text of the reports. Based on your recommendation, we have calculated additional effect sizes for reports where the necessary information is readily available and added this information to the tables.
- We appreciate this feedback and have added brief summaries to the longer subsections of the results reflecting "takeaway" points. The demographics of the samples within this review are largely reflective of the distribution of individuals diagnosed with schizophrenia-spectrum disorders (more frequently male and somewhat older in age). A statement acknowledging the nationality of the samples has been added to the limitations.
- The aim of this review was to provide a descriptive account of the correspondence between subjective and objective reporting and a counterpoint to the view that discrepant reporting is equivalent to invalid data. We have added language to the introduction to further clarify our purpose. Although several of the studies included in the review were already cited as support in our discussion, we have further integrated some of the findings of those reports into the recommendations.
Reviewer 2 Report
Comments and Suggestions for Authors
The paper delves into the crucial aspect of recovery in schizophrenia by examining the incongruity between subjective self-report measures and objective evaluations of cognitive and everyday functioning among individuals with schizophrenia-spectrum disorders. Given the potential implications for treatment and research, the study conducts a systematic review to comprehensively synthesize existing research on this topic, aiming to shed light on the degree of correspondence, or lack thereof, between subjective and objective assessments. In abstract, the last words are bigger than the rest.
Regarding methodology, I consider that the authors could explain better all the steps from PRISMA (Preferred Reporting Items for Systematic Reviews and Meta-Analyses) guidelines, for example: criteria may include population characteristics, interventions, comparators, outcomes, and study designs (PICOS). Moreover, is there a protocol registration? Authors should register the systematic review protocol with a recognized registry (e.g., PROSPERO) before starting the review, and authors should clearly state any deviations from the registered protocol in the final review.
In addition, with all the heterogeneity that exists between the articles included, is a systematic review or a scoping review more advantageous?
The results show weak to moderate relationships between objective evaluations of these areas and subjective self-reports of everyday functioning and cognition. Interestingly, when contrasted to objective measures, people with schizophrenia tend to overestimate their functioning. The review identifies variables that affect the degree of correspondence, and it predicts that there would be more alignment between subjective and objective assessments in cases of depression and better cognitive capacity. Furthermore, it is important to consider the heterogeneity of the schizophrenia spectrum since disparities between subjective and objective assessments may be influenced by a variety of symptom patterns.
There are some typos in the bibliographical references - see, for example, the first reference in the list.
Author Response
- This review was originally designed following PRISMA guidelines prior to the 2020/2021 update. An effort has been made to include updated PRISMA recommendations where possible. It was not pre-registered prior to beginning data collection. A PRISMA statement and statement addressing registration has been added to the manuscript.
- We recognize there is heterogeneity among the reports in this review and were limited in our ability to perform meta-analyses for this reason. However, we feel that the relatively limited scope of our review question along with the methodology used better fits the characteristics of a systematic review. If the reviewers feel strongly that the term scoping review is preferred, we would be amenable to changing this.
- We agree with the reviewer that heterogeneity in symptom presentations may influence concordance of subjective and objective reports. We have included references in our original manuscript's discussion section where some symptoms dimensions have been proposed as possible moderators, but have not been widely studied. We report the relevance of cognitive ability and depression because these factors have more extensive evidence at this time.